# The Impact of Agricultural Credit on the Growth of the Agricultural Sector in Angola

**Mario Augusto Caetano Joao [1,\*] and Abreu Monteiro de Castro [2]**

[1] Department of Economics, Faculty of Economics and Management, Czech University of Life Sciences in Prague, 165 00 Prague, Czech Republic

[2] Department of Economics, Faculty of Economics and Management, Angola Catholic University in Luanda, Luanda 2064, Angola; 1000010815@ucan.edu

\* Correspondence: caetano_joao@pef.czu.cz

**Abstract:** The ultimate goal of this paper was to examine the degree of elasticity between two variables, namely, agricultural credit and agricultural growth, in Angola in the period 2003–2022. Time series data were fitted into the ARDL test using various econometric techniques such as the ADF stationarity test, the Granger causality test, and the ordinary least squares method as well as a vector error correction model (VECM) to analyze the relationship between agricultural credit and agricultural economic growth, showing a causal relationship. Both the impacts through elasticities and the optimal point existing in this relationship were estimated. It was concluded that the impact of agricultural credit on agricultural GDP was 14.41%. The Granger causality test showed signs of a positive linkage between agricultural credit and agricultural GDP. However, there is a causal relationship between agricultural credit and agricultural GDP, in a unidirectional aspect. This result is consistent with most of the earlier studies reviewed in the literature, confirming that credit-oriented monetary policies can boost economic growth and, consequently, development in Angola. It is important for agricultural credit systems to be designed in a way that ensures equitable access, fair interest rates, and appropriate risk management mechanisms. Additionally, monitoring and evaluation mechanisms should be in place to assess the environmental and social impacts of credit programs on agricultural sustainability. It is worth noting that this is a first-of-its-kind study on the matter of the Angolan credit experience, specifically for the agricultural sector. Angola is still searching for a sustainable credit model that could be used as a catalyzer to boost growth and contribute to economic development.

**Keywords:** agricultural sector; agricultural credit; economic growth; Angola; ARDL model; econometric analysis





## 1. Introduction

Agriculture has played a significant role in the history of Angola, one of the largest countries in Africa. The nation's agriculture has evolved over time, influenced by various factors such as colonial rule, independence, civil war, and post-war reconstruction efforts.

Before the arrival of Portuguese colonizers in the 15th century, Angola had a diverse agricultural system. During the Portuguese colonization in Angola, from the late 15th century until 1975, the Portuguese introduced cash crops for export, primarily coffee, cotton, and later on, palm oil, rubber, and sisal.

After achieving independence from Portugal in 1975, the country soon plunged into a civil war that lasted for nearly three decades, until 2002. The conflict severely disrupted agricultural activities, as many farmers were displaced or killed, and infrastructure was destroyed. Agricultural production declined significantly during this period, exacerbating food shortages and increasing dependence on food aid.

Meanwhile, Angola suffered from the so-called Dutch disease, using its oil sector as its main supplier of hard currency to import more than 90% of its consumer goods and services.

Angola's economy is still overwhelmingly driven by its oil sector, being the first oil producer in Sub-Saharan Africa to produce 1.12 million barrels per day (bpd). The country's oil sector accounts for approximately 28% of its GDP, 85% of its overall exports, and 65% of its overall tax revenue. Since 2018, Angola has been changing this paradigm, focusing on its comparative advantages, and especially on re-launching its agribusiness sector.

However, since 2003, Angola has embarked on a process of post-war reconstruction and agricultural reform. The government prioritized revitalizing the agricultural sector to promote food security, economic development, and poverty reduction. Efforts were made to rehabilitate infrastructure, provide technical assistance to farmers, and improve access to credit.

In this regard, the background aim of this study is to evaluate possible drivers of agricultural growth, with the focus in this work being access to credit in agriculture. Banking and financing instruments and products are seen as a catalyst for increasing productivity in the production sector and, in particular, the agricultural sector. Three main factors impact agricultural growth in Angola, namely, the business environment, farmers' knowledge, and the financial system's financing appetite. This study identifies the impact of agricultural credit on agricultural growth in Angola in the period 2003–2022. In particular, the study aims to accomplish the following:

(1) Assess the factor of credit in the agricultural sector;
(2) Critically evaluate the impact of credit on the growth of the agricultural sector;
(3) Provide quantitative results regarding the impact of agri-credit on the growth of the agricultural sector;
(4) Provide recommendations on the importance of credit for the growth of the agricultural sector;
(5) Moreover, the study addresses the following research questions:

- Is credit provided to the agricultural sector a determinant of growth in the sector?
- What is the impact of providing credit to the agricultural sector?
- Should the amount of credit provided to the agricultural sector be increased?

This study is significant as it attempts to identify the impact of credit products provided to Angola's agricultural sector on the growth of the sector from 2003 to 2022. In the research, preliminary analysis showed that there was a direct relationship between GDP and credit. Therefore, this relationship was examined to obtain quantitative results regarding the importance of agricultural credit to the growth of the agricultural sector in Angola and to provide policy recommendations. On the other hand, this is a first-of-its-kind study on the matter of the Angolan credit experience, specifically for the agricultural sector. Angola is still searching for a sustainable credit model that could be used as a catalyzer to boost growth and contribute to economic development.

The rest of this paper is structured as follows: Section 2 describes the information on the significance of the agricultural sector in Angola's economy, Section 3 provides a review of the important literature, Section 4 outlines the methodology used, Section 5 is reserved for the results, and Section 6 provides conclusive remarks.

## 2. Contribution and Potential of Agriculture to Contribute to Angola's GDP

In Angola, the agricultural sector contributed just 6.43% of the total GDP in 2022. Nonetheless, agriculture is one of the biggest and fastest-growing non-oil sectors of Angola's economy.

Agricultural production is currently mostly carried out by smallholder farms, in which approximately three million households (approximately twelve million people, or approximately 30% of Angola's population) are involved. The Angolan government has adopted a national program for the development of the agricultural sector in its National Development Plan 2018–2022, which has two principal objectives: (i) the modernization of smallholdings in order for them to become more market-orientated businesses and (ii) the establishment of large and modern industrial farming businesses. Smallholder farms, on which Angola's agricultural sector is currently based, have much lower production efficiency levels than those in more developed countries.

The credit system in Angola, despite all efforts, remains selective, servicing an exclusive population made up of influential landowners. Furthermore, the system is characterized by high interest rates that involve high-value loans subject to onerous requirements and guarantees, in addition to careful analysis at the time of a loan's concession.

To strengthen the nation's value chain, in 2018, the Government adopted the Program to Support Production, Export Diversification, and Import Substitution (PRODESI), which contains five key initiatives to enhance the diversification of the Angolan economy, with the aim of significantly decreasing its historical over-reliance on oil export revenue and the importation of basic food products.

Having been, during its colonial era, one of the world's largest exporters of coffee and other agricultural commodities such as cotton, sisal, maize, cassava, and bananas, today, Angola has crop and animal production levels far below their potential levels, forcing the country to spend large financial resources on food imports.

The government has identified several key productive areas that can promote investment and foster public–private partnerships, and it set out further initiatives to boost domestic agricultural production to mitigate the country's current expenditure on the importation of basic food products.

Figure 1 shows a graph of the credit provided to the real economy and the agricultural sector from 2003 to 2022. From 2003 to 2005, there was a linear relationship between both economic variables. From 2006 to 2013, less financing flowed to the agricultural sector. From 2014, this trend was reversed, in large part due to public policies, especially the Angola Investe program, which aimed to re-launch the nation's agricultural sector. This trend was later accelerated by the government's clear efforts towards diversifying the country's economy, with the implementation of the PRODESI in 2018 garnering a strong reaction from the banking and private sectors in 2022.

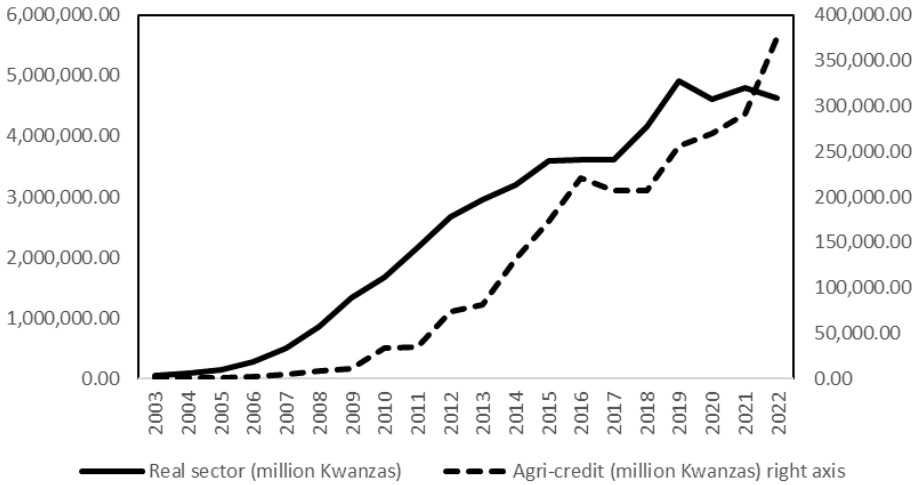

**Figure 1.** Credit provided to real economy and agriculture.

The agricultural sector is characterized by unpredictability in the profitability of its activities and an imbalance in the cash flow of producers due to the sector's dependence on the climate, sanitary conditions, the seasonality of crops, and the cycle of input and market access conditions. Some of this unpredictability is a measurable risk and could be mitigated via information sharing and insurance policies, but a considerable part of it is caused by uncertainty. There are also uncertainties associated with institutional changes in agricultural policies in competing countries and the high volatility of commodity prices. These problems increase transaction costs.

Nevertheless, Figure 2 shows that in Angola, during the period 2010–2022, there was a steadily increasing trend of credit finance provided to the agricultural sector, representing in 2022 a share of 8.11% of the total credit provided to the real sector.

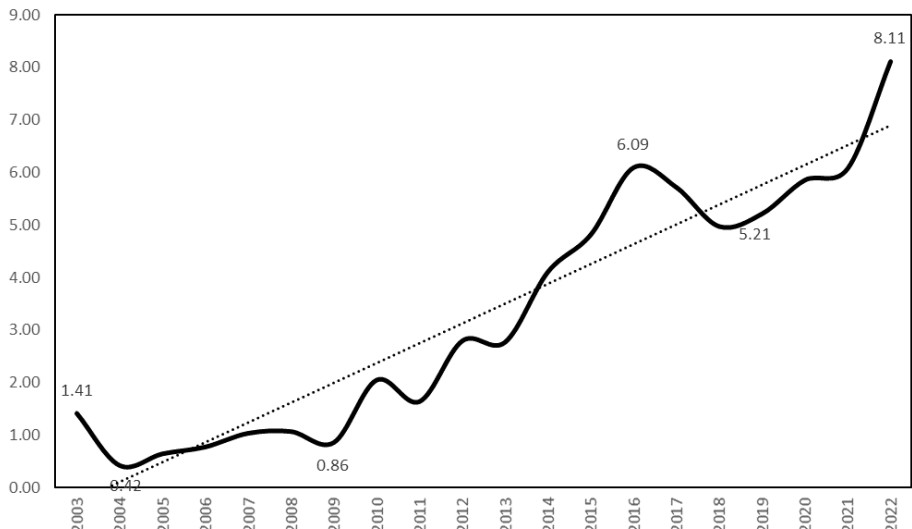

**Figure 2.** Share of agricultural credit in total credit provided to the economy (%).

Over the last 20 years, several policies have been implemented by the government to mitigate risk, including the creation and operationalization of several financial instruments and products at the Angola Development Bank (BDA in Portuguese). Additionally, since 2006, various non-banking financial engineering products have been made available, such as venture capital via the Angolan Venture Capital Active Fund (FACRA in Portuguese) and credit guarantees via the Credit Guarantee Fund (FGC in Portuguese). However, the first concrete steps towards commercial banking resources being broadly available to finance the agricultural sector were made possible with the publication of Notices 4 and 7/19, 10/20, and 10/22 of the National Bank of Angola (BNA). These notices allowed Angola's commercial banks (25 in total) to use the mandatory reserves set by the central bank to provide agricultural credit amounting to a minimum of 2.5% of the total value of a lender's net assets.

Figure 3 shows that, despite some peaks, there was a certain correlation between GDP growth and agricultural GDP growth in the study period, with the agricultural sector being resilient throughout the period and registering stable growth of 5% on average from 2020 to 2022.

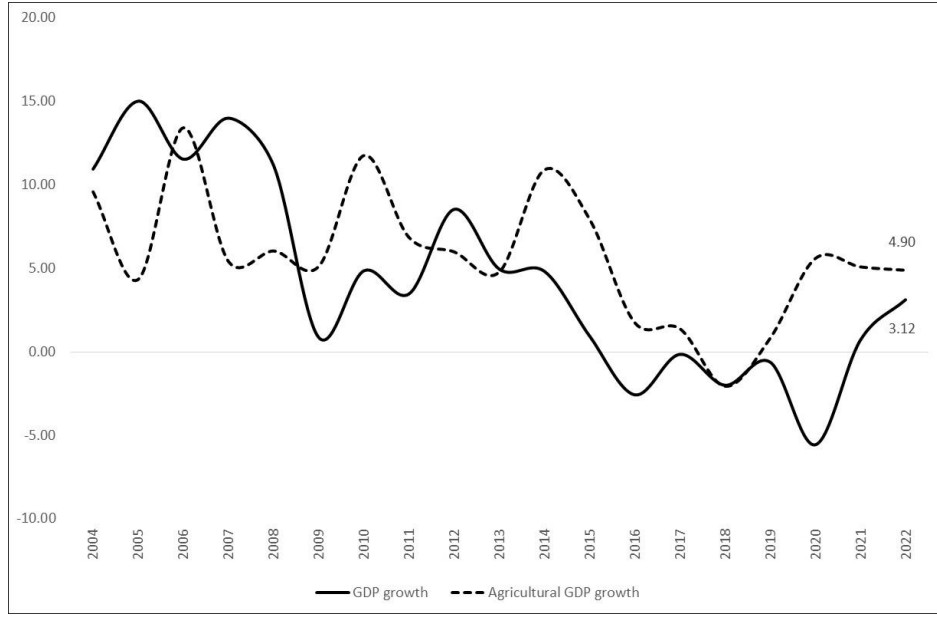

**Figure 3.** GDP growth and agricultural GDP growth rates (%).

Consequently, Figure 4 shows that, in 2022, the agricultural sector accounted for 6.43% of the total gross domestic product (GDP) of Angola; however, its exports were still very limited, representing less than 1% of the country's overall food produced.

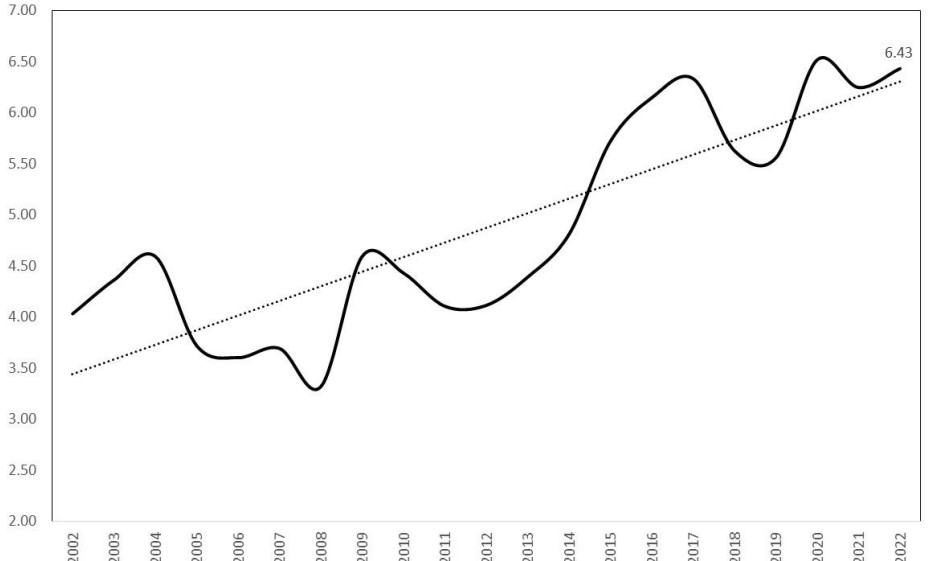

**Figure 4.** Contribution of agriculture to Angola's GDP (%).

By 2022, Angola's commercial banks and the BDA had provided AOA 375.28 billion in credit to the country's agricultural sector, equivalent to approximately USD 750 million, representing an 82% increase in nominal terms compared with 2018, when the PRODESI was implemented.

## 3. Literature Review

Many theoretical and empirical studies have analyzed the correlation between financial credit and economic growth. Some have found positive impacts and others have found negative impacts of financial credit on economic growth, but the literature's general conclusion is that strong financial systems increase credit activity and, subsequently, lead to economic growth.

Empirical studies such as those conducted by Pistoresi and Venturelli (2012) [1] show that countries with strong financial systems could benefit from sustainable economic growth. The authors developed a dynamic panel using a generalized method of moments to examine how venture capital and direct investment impact economic growth.

King and Levine [2] proposed a model that examines how financial systems can impact economic growth, leading to improved productivity in four ways, namely, (i) refined selection of the best bankable projects; (ii) rationalization of the financial resources used to finance projects; (iii) enabling investors to diversify the risks associated with innovative activities; and (iv) potential compensation provided by financial systems for innovation.

Leitão [3] proposed an endogenous model that analyzes the correlation between financial credit and economic growth by introducing variables such as domestic credit, savings, bilateral trade, and inflation. The findings showed that savings encourage growth, while inflation is negatively correlated with economic growth.

Financing is used in the agricultural sector to support the supply of agricultural inputs and to support the production process, distribution, and marketing of agricultural products. The demand for agricultural finance is very high. Most smallholders and agricultural SMEs find it difficult to access finance and thus engage in agricultural practices that result in dismal agricultural yields [4].

Castro and Teixeira [5] analyzed the positive impact of interest rate equalization (ETJ) for smallholders and commercial agriculture on GDP growth through increases in the

collection of taxes, comparing it with the amount spent on ETJ. Additionally, Castro and Teixeira [6] measured the elasticity of demand for inputs in relation to rural credit in Brazil and found a very strong elasticity of 0.95 for fertilizers.

The results of using the different econometric models, including autoregressive distributed lag (ADL or ARDL), showed that there was a positive effect of agricultural credit on economic growth [7]. Further, Akram et al. [8] concluded not only that was there a positive effect of agricultural credit on agricultural GDP in Pakistan but also that there was an elasticity of agricultural credit in relation to poverty of −0.35% and −0.27% in the short term and long term, respectively.

Another important study that employed the VAR methodology was conducted by Melo et al. [9], which concluded that a 1.9% increase in credit provided to agriculture generated an impact of 0.79% on agricultural GDP.

On the other hand, some authors [10,11] argue that the financial system–economic growth relationship is not significant. Koivu [12] analyzed the correlation between economic growth and credit performance in the private sectors of 25 countries in transition from 1993 to 2000 and found that decreases in GDP caused by large amounts of credit were mostly related to counterproductive investments.

Pham and Nguyen [13] studied the impact of credit on economic growth in Vietnam with the use of the ARDL model as the analysis method. Their results showed that there was a relationship between credit and GDP, but its expansion in the long term had a negative impact on economic growth in Vietnam. The study proposed a couple of recommendations to avoid the negative impact of credit on the economy, such as better central bank supervision.

On the other hand, Kaleemuddin and Masih [14] also employed the ARDL model to examine the relationship between financial development and economic growth in India. In this case, in line with King and Levine [2], the study findings show that financial development and economic growth are cointegrated in the long run, meaning that reforms in the financial banking and non-banking sectors can increase economic growth.

FitzGerald [15] studied institutional structures as important catalyzers of the financial sector's impact on economic growth, especially via fiscal and monetary policy measures that should provide low and stable real interest rates and competitive exchange rates, as well as appropriate tax incentives.

Notably, all the above authors agreed on the positive impact of agri-credit on the formation of agricultural GDP. With regard to the development of the present study, the work of Borges and Parré [16] served as a particularly important reference.

## 4. Methodology

An econometric data model was chosen to help define the inter-relationships between variables in order to use the model as a catalyzer for long-term sustainable development. Additionally, it also serves to compare with the preliminary analysis of the relationship between the two variables in Section 2.

### 4.1. Data Gathering

The time series data were extracted from the Angolan Central Bank (BNA: Agricultural Credit Statistical Yearbook) and Angola Statistical Office (INE: Quarterly National Accounts). The time horizon is an annual series, covering the period 2003–2022, making a total of 20 observations. The determination or choice of this period was justified by the availability of data and political stability that occurred in Angola as a result of the peace achieved after 27 years of post-independence civil war (1975 to 2002). The political regime and party in power from 2003 to 2022 were the same.

It is worth noting that, throughout the colonial period, Angola was among the largest exporters of agricultural products worldwide. The recovery of this profile in the global agricultural sphere and the achievement of food security requires a lot of effort both from the government and the private sector.

To estimate elasticities, agricultural GDP values were chosen on an annual basis and, in real terms, adopted as the dependent (explained) variable of the model to be estimated. The independent (explanatory) variable used was the total agricultural credit. Thus, it is expected that the total agricultural credit, in this estimation, positively impacts agricultural GDP.

It is important to highlight that the purpose of this study is to evaluate the relationship between total agricultural credit and agricultural GDP in the analyzed period, through robust methods, with the estimation of an econometric model using the ARDL test.

In order to address the research aims or questions, respectively, we used the reduced form of the transmission mechanism of monetary policy in a credit channel with a focus on the agricultural sector. Moreover, we used the Keynesian assumption about the exogeneity of the transmission mechanism. That is, the economic model represents a functional relationship between Angola's GDP as a dependent variable and agricultural credit as an independent variable.

To examine the association between the two variables (agricultural credit and agricultural GDP), the following hypotheses were put forward:

**H0.** *Agricultural credits have no significant impact on the agricultural GDP in Angola;*

**H1.** *Agricultural credits have a significant impact on the growth of the agricultural GDP in Angola.*

### 4.2. Econometric Model

The ARDL model has become an important tool for detecting the cointegration relationship based on the work of Pesaran and Shin [17]. The authors demonstrate that with an ARDL representation, it is possible to identify cointegration relationships in a system formed by variables that are all I(1), all I(0), or a mixture of stationary variables and variables I(1). This constitutes a great advantage compared with the Johansen cointegration method and even the FMOLS estimator, as both assume that all variables in the system are I(1).

For this reason, in the case of long-term time series data for an economy with large probabilities of lag, we believe the choice to be better suited, that is, to test the model's estimators/regressors using the ARDL test.

The dynamics of the relationship between GDP and agricultural credit are analyzed using the ARDL model given the agricultural credit exogeneity assumption. That is, ARDL(p,k) can be written as:

$$Y_t = \alpha_0 + \sum_{i=1}^{p} a_i Y_{t-i} + \sum_{i=1}^{p} b_i X_{t-i} + \varepsilon_{1t} \tag{1}$$

where
  $Y_t$ is the endogenous variable (GDP);
  $Y_{t-i}$ represents the lagged values of the endogenous variable;
  $X_t$ is the exogenous variable included in the model (agricultural credit);
  $X_{t-i}$ represents the lagged values of the exogenous variable;
  $\alpha$ is the parameter;
  $\varepsilon_{1t}$ represents the error terms of the model.

According to Borges and Parré [16], it is necessary to verify the degree of stationarity of the series used in the ARDL model. The Dickey and Fuller [18] test is used to determine the level of integration of the series and to consider the possible differences that make the series stationary.

According to Borges and Parré, if the series is stationary with respect to 0 (zero), it will be conducted to the point where it is estimated as in Equation (1). If the series is integrated into Equation (1), the equation uses the variables according to the first difference.

Moreover, we used the Granger test to investigate the causality between GDP and agricultural credit. Enders [19] argues that the Granger test proposes testable definitions of the causality of two time series, based on the assumption that the cause precedes the effect. In this case, the test's objectivity is restricted in the sense of observing how much of

variable *Y* is explained by the values of variable *X*, that is, determining if variable *X* can explain the observed value of *Y*, meaning that it determines whether the values of *X* are statistically significant.

To assess the stationarity of the time series in the present study, the augmented ADF test was used; the purpose of this test is to guide the level of integration between series by measuring the number of differences required for a series to be stationary.

### 4.2.1. Dickey–Fuller (ADF) Test

For the analysis of the stationarity accuracy of the estimation model, it is common to use the augmented Dickey–Fuller (ADF) test since it is a single-root test that allows econometrical identification in time series of the significant existence of variables showing a stochastic trend through a hypothesis test; this is assessed using the following equation:

$$Y_t = \alpha + \delta Y_{t-1} + \varepsilon_t$$

The augmented Dickey–Fuller test suggests three different forms under three different null hypotheses. For each case, the null hypothesis is $\delta = 0$ and the series has a unit root and is nonstationary:

(1) $Y_t$ is Random Walk: $\Delta Y_t = \delta Y_{t-1} + \varepsilon_t$
(2) $Y_t$ is Random Walk with drift: $\Delta Y_t = \beta_1 + \delta Y_{t-1} + \varepsilon_t$
(3) $Y_t$ is Random Walk with drift and trend: $\Delta Y_t = \beta_1 + \beta_2 t + \delta Y_{t-1} + \varepsilon_t$.

### Dickey–Fuller Test for Unit Root Test

The tables below (Tables 1–4) present the augmented ADF test of the time series. The results of the augmented ADF test in the first difference show that the TOTAL_AGR_CR series is I(0) stationary at a 5% level of significance. GDP_AGR was nonstationary at the same significance level. However, it was possible to make the GDP_AGR series stationary in the first difference. This was achieved by generating the first difference variable and running the unit root test one more time.

**Table 1.** Nonconstant lags (1).

| Variable | ADF (1) | Critical Value | | |
|---|---|---|---|---|
| | Noconstant Lags | 1% | 5% | 10% |
| GDP_AGR. | −0.065 | −2.660 | −1.950 | −1.600 |
| TOTAL_AGR_CR. | 2.798 | −2.660 | −1.950 | −1.600 |

**Table 2.** Trend lags (1).

| Variable | ADF (1) | Critical Value | | |
|---|---|---|---|---|
| | Trend Lags | 1% | 5% | 10% |
| GDP_AGR. | −3.855 | −4.380 | −3.600 | −3.240 |
| TOTAL_AGR_CR. | −1.279 | −4.380 | −3.600 | −3.240 |

**Table 3.** Drift lags (1).

| Variable | ADF (1) | Critical Value | | |
|---|---|---|---|---|
| | Drift Lags | 1% | 5% | 10% |
| GDP_AGR. | −1.463 | −2.602 | −1.753 | −1.341 |
| TOTAL_AGR_CR | 1.821 | −2.602 | −1.753 | −1.341 |

It is possible to check in Table 1 the stationarity of the series, from the nonconstant lags perspective, the results demonstrate that the TOTAL_AGR_CR series, show I(0), stationary at a level of up to 10% significance. However, the GDP_AGR series was non-stationary at

this level, but became stationary in first difference, being integrated of first order I(1), as can be seen in Table 4.

**Table 4.** Generate (nonconstant lags (1)).

| Variable | ADF (1) | Critical Value | | |
|---|---|---|---|---|
| | Generate (Noconstant Lags) | 1% | 5% | 10% |
| GDP_AGR. | −5.145 | −2.660 | −1.950 | −1.600 |
| TOTAL_AGR_CR | −0.511 | −2.660 | −1.950 | −1.600 |

Source: All prepared by the authors.

The trend of series in the ADF test, as shown in Table 2, demonstrates that the serie GDP_AGR is I(0), stationary between 5% and 10% significance. The TOTAL_AGR_CR series presents a non-stationary trend.

The information in Table 3 illustrates that the derivative of the series in the ADF test is I(0), stationary at 10% significance for GDP_AGR and between the 5% and 10% significance level for TOTAL_AGR_CR, respectively. This means that the null hypothesis is rejected, with the series of the model having a stationary variation at the 10% level of significance.

Regarding the information contained in Table 4 and given the fact that the GDP_AGR series does not show I(0), it was non-stationary according to the data in Table 1, but is was possible to become stationary in the first difference, being integrated at first order I(1), as is visible.

It is common to represent the null hypothesis for the augmented ADF test analysis as follows (https://pt.economy-pedia.com/11038607-dickey-fuller-test, accessed on 23 March 2022):

**ADF H$_0$.** $\delta = 1$ *Stochastic trends in time series.*

**ADF H$_1$.** $\delta < 1$ *Absence of stochastic trends in time series.*

In the augmented ADF test, the null hypothesis reflects the possibility of nonstationary data, which implies the presence of a unit root in the variables of the estimated model, and the alternative hypothesis is that the data are stationary. However, the following rule exists: if the t-statistic is greater than the critical 5% value, the null hypothesis is rejected. Therefore, the data are stationary. However, if the t-statistic is less than the critical value of 5%, the null hypothesis is accepted. Therefore, the data are nonstationary. In order to correct the variables (of the model), the first difference variables are generated and the unit root test is run once more.

### 4.2.2. Autoregressive Distributed Lag Model Estimation

Considering a linear model for time series:

$$Y_t = \alpha + \varnothing Y_{t-1} + D(L)X_t + \varepsilon_t$$

where $D(L) = \delta_0 + \delta_1 L + \ldots + \delta_s L^s$, with $L$ representing the lag operator, that is $LX_t = X_{t-1}; L^2 X_t = X_{t-2}; \ldots + L^s = X_{t-s}$, where $X_t$ represents the set of explanatory variables of the model and $\varepsilon_t$ is the error term. This is the most explicit version of an ARDL model, for which the stability condition is assumed to be satisfied, $|\varnothing| < 1$; however, there are unit root problems, although they are not very predictable in practice.

The same author outlines that among the strong points of this model is its generality, that is, because it has a more conservative perspective, the articular cases that come from this model are very extensive, ranging from the static regressions themselves, without dynamics ($\varnothing = 0$ $e$ $\delta_1 = \delta_2 = \ldots = \delta_s = 0$), until we reach the VECM.

The Vector Error Correction Model contains a variety of applicability in economic series given its components that make it a flexible model. Much of this is on account of it being a model capable of incorporating both short- and long-term mechanisms, as well as the speed of adjustment with which balance is re-established. In general, the VECM is a

particular case of the ARDL model; to be more concise, an ARDL (1,1) model with only a single explanatory variable is expressed, as is the case in the present study:

$$Y_{1t} = \mu + \alpha_1 Y_{1t-1} + \beta_0 Y_{2t} + \beta_1 Y_{2t-1} + \varepsilon_t$$

which results in,

$$\Delta Y_{1t} = (\alpha_1 - 1) + \left[ Y_{1t-1} - \frac{\mu}{1-\alpha_1} - \frac{\beta_0 + \beta_1}{1-\alpha_1} Y_{2t-1} \right] + \beta_0 \Delta Y_{2t} + \varepsilon_t$$

Looking at the previous equation, Aparício describes four characteristics of the ECM:

(a) It is a short-term dynamic model since, if there are no imbalances, $\Delta Y_{1t}$ is driven only by $\Delta Y_{2t}$;

(b) The existing cointegration is incorporated into the model ($Y_{1t-1} - \frac{\mu}{1-\alpha_1} - \frac{\beta_0+\beta_1}{1-\alpha_1} Y_{2t-1}$) and these quantities are specified in the model;

(c) Therefore, one can estimate the speed with which equilibrium is re-established through ($\alpha_1 - 1$);

(d) Finally, the long-term multiplier is expressed by $\frac{\beta_0+\beta_1}{1-\alpha_1}$, and, on the other hand, the short-term multiplier is given by $\beta_0$ due to the fact that quantities are specified in the model.

There is the issue of specification in the VECM model when estimating the regression parameters, that is, the coefficients of the explanatory variables are seen as a problem in the model, knowing that the OLS estimator is BLUE (Best Linear Unbiased Estimator). On the other hand, to guarantee the satisfaction of this and other hypotheses necessary to validate the statistical inference and validity of the final chosen model, several tests are performed, such as the Breusch–Godfrey test, Durbin–Watson test, etc. Therefore, for the present study, we chose to perform the Durbin–Watson test.

There is an autocorrelation problem in the estimated model when the errors or residuals of the times series are correlated. Therefore, in order to diagnose autocorrelation problems in the time series studied, the Durbin–Watson (DW) t statistic was used, followed by an analysis of the rule or decision diagram. In order to solve possible problems of autocorrelation in the model series, Prais estimation was used, and with this, the consistency of the predictability of the estimated econometric model was maintained.

## 5. Results

Preliminary analysis using the simple regression model, as shown in Figure 5, found that there was a direct relationship between GDP and credit provided to agriculture, that is, every monetary unit resulted in an increase in product units.

In terms of presentation, Figure 5 demonstrates the simple linear regression econometric model, which has the following form:

$$\hat{Y}_i = 556.4833 + 14.41205 X_i$$
$$(636, 227.4) \quad (3.57562)$$
$$R^2 = 0.4744$$

where the values in parentheses represent the standard deviations of the estimated econometric model parameters.

As shown by the results, a very positive impact of total agricultural credit on the formation of agricultural GDP was found. This strong correlation allows for solid sustainable policies in the sense that every monetary unit of kwanza would result, ceteris paribus (keeping everything else constant), in a 14.41205 unit increase in agricultural GDP. However, this approach omits the dynamic relationship between agricultural GDP and agricultural credit which can be assumed to be highly relevant. That is why we use dynamic model specification in the form of the ARDL model.

Table 5 presents the results of the estimated econometric model. Demonstrates the positive relationship between the coefficients of the model variables. From the time se-

ries analyzed, we were able to notice the impact of the TOTAL_AGR_CR variable on GDP_AGR. With a 1% variation in TOTAL_AGR_CR causing a positive variation of 14.41% in GDP_AGR.

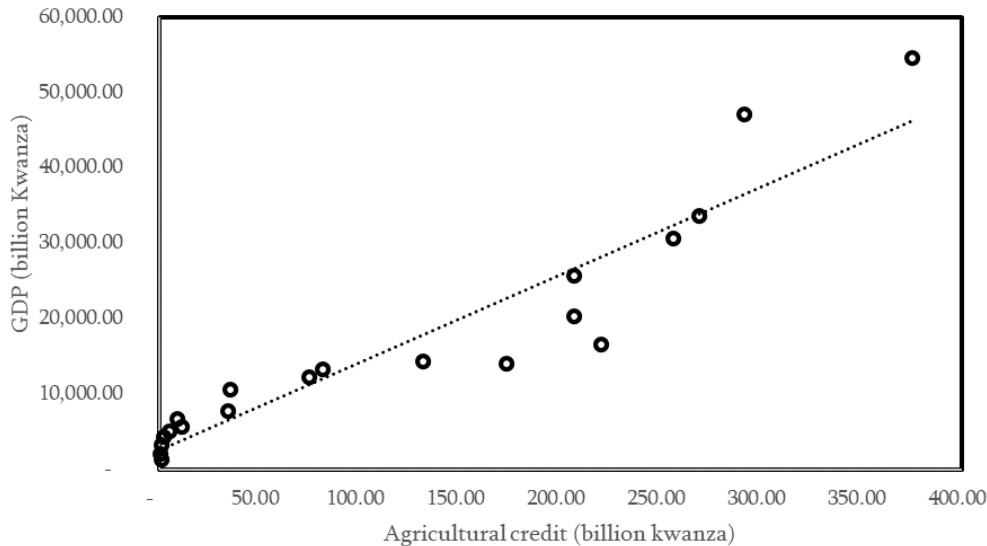

**Figure 5.** The evolution of GDP and agricultural credit.

**Table 5.** Estimated.

| Variable and Test | Coefficient | Standard Error | *t* Statistic | *p*-Value |
|---|---|---|---|---|
| TOTAL_AGR_CR | 14.41205 | 3.57562 | 4.03 | 0.001 |
| Constant | 556.4833 | 636,227.4 | 8.75 | 0.000 |
| Number of observ. | 20 | - | - | - |
| F test | 16.25 | - | - | - |
| Prob > F | - | - | - | 0.0008 |
| R-squared | 0.4744 | - | - | - |
| Adj R-squared | 0.4452 | - | - | - |

It is important to highlight, according to the information in Table 6, that the time series of the model for the period under analysis in this study showed autocorrelation problems. Therefore, the Prais test was necessary, which made it possible for the elasticities presented to comply with the expected signs.

**Table 6.** Prais (error correction of model).

| Variable and Test | Coefficient | Standard Error | *t* Statistic | *p*-Value |
|---|---|---|---|---|
| TOTAL_AGR_CR | 14.93363 | 2.274437 | 6.57 | 0.000 |
| Constant | 551.2012 | 393,115.8 | 14.02 | 0.000 |
| Number of observ. | 20 | - | - | - |
| F test | 51.86 | - | - | - |
| Prob > F | - | - | - | 0.0000 |
| R-squared | 0.7423 | - | - | - |
| Adj R-squared | 0.7280 | - | - | - |
| Durbin-Watson (original) | 2.876943 | - | - | - |
| Durbin-Watson (transformed) | 2.172601 | - | - | - |

Source: All prepared by the authors based on research data.

As can be seen in the result of the DW test, the estimator of the time series estimated in the present study presents a test DW = 2.876943, with k = 2 and *n* = 20, where k represents the parameters and *n* represents the number of observations used in the model.

Analyzing the values after Prais estimation, it can clearly be verified that there is a new value for the Durbin–Watson (DW) test = 2.172601.

The acceptance or rejection of the null hypothesis of a model largely depends on the choice of the level of significance, that is, on the value of $\alpha$. For the model estimated in the present study, the level of significance ($\alpha$) is equal to 5%.

Thus, analyzing the values extracted from the results of the estimated econometric model, the null hypothesis was found to be rejected because the $\rho$ value (P > |t|) of the model parameters was lower than the significance level, which implies that the parameters of the present estimated model were statistically significant.

Further, it was observed that the $\rho$ value (Prob > F) of the F statistic was lower than the significance level, implying that the econometric model estimated in the present study was globally significant.

## 6. Conclusions

This study attempted to determine the importance of agricultural credit to the overall performance of agricultural production in Angola.

According to the values of the estimated econometric model, a positive relationship was found between agricultural credit and agricultural GDP. In other words, agricultural credit has played a major role in the formation of agricultural GDP in Angola, and above all, in its progressive growth.

Clearly, the government's current attempts to induce the banking sector to provide more credit to the agricultural sector need to continue and potentially even accelerate, which could be achieved by making more financial resources available. The government's implementation of different initiatives that provide negative real interest rates (subsidized interest rates) through banking financial instruments and products via the Angola Development Bank (BDA), or programs with local commercial banks, such as the Angola Investe program (Investing in Angola), BNA Notice 10/2022 and, most recently, the PAC (Credit Support Project), along with the technical assistance provided by the PRODESI, has brought a significant shift in the country's agricultural sector's performance.

In July 2022, the Government of Angola approved an ambitious and massive grain production plan called PLANAGRÃO (in Portuguese, Plano Nacional de Fomento para a Produção de Grãos). The government will secure an approximately USD 4 billion financial package via the Angola Development Bank aimed at inducing the private sector to produce four priority grains in the time period 2023–2027, namely, maize, rice, wheat, and soya, with a view to reducing the country's dependence on food imports and ensuring its self-sufficiency and food security. The agricultural credit used to implement PLANAGRÃO might have a significant impact on the agricultural GDP of the period.

However, ensuring such initiatives are sustainable for their main objective (users) involves a much broader approach and does not end only with their implementation. Continuous monitoring is of the utmost importance to ensure that the financial credit granted is really applied to the development of agricultural activities and is not used elsewhere. Most agricultural agents have expressed enormous concerns about the transparency of access to credit, as a large number of farmers who truly produce and who need it the most to improve their productivity have often been excluded from this process.

In this study, a literature review found that most of the studies assessed reported a positive impact of agricultural credit on the formation of agricultural GDP, with similar findings to those of King and Levine [2] and Kaleemuddin and Masih [14], the latter also employing the ARDL model. Further, the study found that a variation of about 1% in agricultural credit would result in a positive variation in agricultural GDP of around 14.41%. Therefore, agricultural credit will, in principle, serve to expand the demand for agricultural

goods, although this does depend on the expression of agricultural credit in the formation of agricultural GDP.

This study aims to define the impact of agricultural credit on agricultural GDP, which is closely related to sustainability in several ways. First, enhancing agricultural productivity: access to credit allows farmers to invest in modern agricultural practices, such as improved seeds, fertilizers, irrigation systems, and machinery. These investments can lead to increased productivity, higher crop yields, and more efficient resource utilization. Second, promoting sustainable farming practices: credit can be used to fund the adoption of sustainable farming techniques. For example, farmers can utilize credit to implement organic farming methods, integrate agroforestry systems, practice integrated pest management, or invest in renewable energy solutions for their farms. Third, encouraging investment in agri-businesses: credit can facilitate the growth of agri-businesses and value chains, such as storage facilities, processing plants, and distribution networks. These investments can help reduce post-harvest losses, improve food quality and safety, and enhance market access for farmers.

Although there are several studies related to the history of the Angolan economy, its economic growth, financial system, credit models, etc., it is worth noting that this is a first-of-its-kind study on the matter of the Angolan credit experience, specifically for the agricultural sector. Angola is still searching for a sustainable credit model that could be used as a catalyzer to boost growth and contribute to economic development.

Finally, this study opens up different avenues for future scientific research regarding the state of the art of the agricultural sector credit and GDP, using different econometric model approaches.

## 7. Limitations

One of the main limitations of this work is that, as data for a longer series were not available, as well as the recognition of structural breaks, there may have been unexplained variability in the data sample; that is, statistical noise may not have been minimized as much as it could have been. Further, it is worth noting that agricultural products also depend on variables other than agricultural credit, such as infrastructure (especially roads destined for agricultural exploration areas and the connection of these with shopping centers), utilities, technical assistance, scientific studies, etc.

**Author Contributions:** Conceptualization: M.A.C.J.; methodology: A.M.d.C.; software: A.M.d.C.; validation: M.A.C.J. and A.M.d.C.; formal analysis: M.A.C.J.; investigation: M.A.C.J. and A.M.d.C.; resources: M.A.C.J.; data curation: A.M.d.C.; writing—original draft preparation: M.A.C.J.; writing—review and editing: M.A.C.J. and A.M.d.C.; visualization: M.A.C.J.; supervision: M.A.C.J.; project administration: M.A.C.J. All authors have read and agreed to the published version of the manuscript.

**Funding:** This research received no external funding.

**Data Availability Statement:** The primary data used in this research can be found in the Angola National Accounts published by the Angola Statistical Office (www.ine.gov.ao, accessed on 12 June 2021), and the credit data are published by the Angola National Bank (www.bna.ao, accessed on 12 June 2021).

**Conflicts of Interest:** The authors declare no conflict of interest.

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
