# Peer review of "The Impact of Agricultural Credit on the Growth of the Agricultural Sector in Angola"

_sustainability, doi:10.3390/su152014704_

Round 1
Reviewer 1 Report
1. This is a very important research for agricultural economy, this research has good practicability and contribution.
2. The literature review in Section 3 can add related research using the ADL model.
3. You can explain the reasons for choosing the ADL mode and its appropriateness for this study.
4. In the conclusion, you can explain the differences or similarities between using this method and previous literature.
Author Response
2. The literature review in Section 3 can add related research using the ADL model.
R: We have added in chapter 3 more detail on the literature used, especially paragrapgh 6 (Moura), and added paragpahs 9 and 10 with two more literature references (Pham and Nguyen (2020) and Kaleemuddin and Masih (2017)).
3. You can explain the reasons for choosing the ADL mode and its appropriateness for this study.
R: reasons explained with paragraphs 1 and 2 of the subchapter 4.1 on econometric model.
The ARDL model has become an important tool for detecting cointegration relationship based on the work of Pesaran and Shin (1999). The authors demonstrate that with an ARDL representation, it is possible to identify cointegration relationships in a system formed by variables that are all I(1), all I(0), or a mixture of stationary variables and variables I(1). This constitutes a great advantage compared to the Johansen cointegration method and even the FMOLS estimator, as both assume tha all variables in the system are I(1).
4. In the conclusion, you can explain the differences or similarities between using this method and previous literature.
R: in paragraph 6 of the conclusions, we have explained. In this study, a literature review found that most studies assessed reported a positive impact of agricultural credit on the formation of agricultural GDP, with particular similar findings with King and Levine (1993) and Kaleemuddin and Masih (2017), this latter also employed the ARDL model. It was found that most of them are in agreement regarding the positive impact that agricultural credit has on the growth of agricultural GDP. Therefore, in terms of analysis between the present study and previous literature regarding the method used, a comparative method was established with criteria that described the Differences: in the explanatory variables and model estimation test; as for Similarities: object of study, explained variable, Granger causality and results found. Overall, the present study is in line with the vast majority of research presented in the literature review section.

Reviewer 2 Report
Dear Authors, thank you very much for an interesting and important article in the field of research on the relationship between agricultural credit and GDP growth. It should be noted the qualitative elaboration of the article, research methodology, setting goals, objectives, research questions, hypotheses. In order to make the material more valuable for readers, I propose to make a small clarification in several places. Please complete the methodological principles and approaches of the study, the rationale for choosing data for modeling. It would be interesting to analyze the regulatory framework for the impact of agricultural lending on the growth of agricultural production in foreign countries. In conclusion, it is necessary to note the scientific novelty and practical significance of the results of the study. I hope my suggestions will improve the text.
Author Response
Dear Authors, thank you very much for an interesting and important article in the field of research on the relationship between agricultural credit and GDP growth. It should be noted the qualitative elaboration of the article, research methodology, setting goals, objectives, research questions, hypotheses. In order to make the material more valuable for readers, I propose to make a small clarification in several places. Please complete the methodological principles and approaches of the study, the rationale for choosing data for modeling. It would be interesting to analyze the regulatory framework for the impact of agricultural lending on the growth of agricultural production in foreign countries. In conclusion, it is necessary to note the scientific novelty and practical significance of the results of the study. I hope my suggestions will improve the text.
R: We have added a new paragraph in chapter 4 (Methodology) explaining the rationale for chosing data modeling, making clear the data origins, time series. In clonclusion, we have added in the last two paragraphs the novelty, explaining that although there are several studies related to the history of the Angolan economy, its economic growth, financial system, credit models, etc., it is worth noting, that this is the first of a kind study on this matter on the Angolan credit experience, specifically for the agricultural sector. Angola is still searching for sustainable credit model that could be used as a catalyzer to boost growth and contribute to economic development.
Additionally, we have adjusted the abstarct in the beginning of the article.

Reviewer 3 Report
Dear authors,
After reading critically the submitted manuscript I’m listing some comments, questions, doubts, and suggestions that I firmly believe can improve your work substantially. The work aims to examine the degree of elasticity of agricultural credit and agricultural growth, in Angola in the period between 2003 and 2022, a very interesting topic in terms of impacting the agricultural growth through the financial channel.
1. From the structural point of view of the manuscript I can observe and suggest:
· The abstract lacks the mention of the methodological approach of the work, and the main results are not explained clearly. I observed the last phrase: “The outcome of this analysis indicated that agricultural credits have a significant impact on agricultural growth.” which seems to formulate one finding, too much general and kind of a fact one can find through the literature (not any novelty here!). My suggestion here: try to crystalize your work in your abstract in the best way in order for the reader to get the essence of the analysis.
· At the introduction part: the section comes to fragmentized, the ones that you numbered as subsections could be considered as broad paragraphs in order to see a connection between the pieces. I suggest reformulating the Introduction part taking into account the most important thing here: which is the lacune, the gap in the revised literature that you aim to meet with your work. It should be explicitly stated, underlining the novelty behind it (if there is any!).
· I suggest considering merging the Introduction part with the Literature review part.
· The methodology section should have a subsection about the data. I missed the type of data and the variables used explained clearly.
· The section on Results should be formulated clearly about your findings. Things such as Table 8 or the note on page 18 are put there for what reason? DW test table values are known from the researchers, so it seems to me unnecessary to be included. It presents the work in a scholastic language style, not too refined in my opinion.
· Statement such as “…Introduce ADF test here…." on page 9, gives me the idea of an unpolished first version of the manuscript.
2. In terms of content:
· The title: two generic (a personal preference of mine).
· The results should be aligned also with the line of the journal. Where are your findings located in terms of sustainability issues?
· The methodological part: I should express that reading the revisited literature and what you stated in the abstract part: “…doubts have been expressed about the accuracy of econometric models that measure the real impact that banking financial institutions have on boosting the agricultural sector…” I was expecting a very consolidated methodology (in statistical and econometrical terms). Your choice about the econometric model ADL (p, k) (page 9 it is written ADL(p, p), also at the equation 1) is not justified. Why not an autoregressive model AR(p), or a distributed lag model (DL(k)) (selection criteria: serial correlation of residuals, information criteria. significance)?
· Then you pass (on page 9 again) from the stationarity attribute of the series to the Granger causality approach and then (page 13) at the ECM approach doubting for cointegration between the two series. Surprisingly I found a simple linear OLS approach on page 14 as the first empirical result. My observation: there is not a well-designed research analysis. My suggestion: There is a clear empirical path to follow: series stationarity -> cointegration -> (if positive) VECM -> FEVD, IRFs and Granger causality; ->(if negative) VAR -> FEVD, IRFs and Granger causality.
· Moreover: the data is annual, or monthly? I missed it. However, the period of study is a wide period (2003-2022) covering several structural breaks and regime shifts that are not considered. The justification behind this fact? This is not a realistic assumption. All the used techniques (without considering the presence of several structural breaks) provide nonreliable findings and results. More specifically:
(a) There are tests to detect structural changes in the series (first step).
(b) The Augmented Dickey-Fuller (1979) unit root test is not appropriate for structural breaks. It has low power. There are unit root tests to apply in the presence of structural breaks (second step).
(c) It is possible to run a cointegration analysis between series in the presence of structural breaks in the deterministic trend (Johansen et al, 2000) (third step).
· Graphical series representations are a bit confusing, and figures and tables lack clearly formulated captions.
· The work aimed to examine the degree of elasticity of agricultural credit and agricultural growth, in Angola. Nothing about this expression was mentioned or calculated in the Result part. Moreover, are you talking about short or long-run elasticity?
In general, the topic is interesting but limited in terms of used techniques. I firmly the methodological part of your work is the one that needs to be invested more.
My best regards!
The language style needs to be polished and grammatically and typos need to be corrected.
Author Response
Dear Researcher,
First of all, thank you for your didactic-methodological insights which allowed us to rationalize the article to better fit the purpose. We used track changes to better show our reaction.
The abstract lacks the mention of the methodological approach of the work, and the main results are not explained clearly. I observed the last phrase: “The outcome of this analysis indicated that agricultural credits have a significant impact on agricultural growth.” which seems to formulate one finding, too much general and kind of a fact one can find through the literature (not any novelty here!). My suggestion here: try to crystalize your work in your abstract in the best way in order for the reader to get the essence of the analysis.
R: the abstract has been substantially adjusted to reflect these gaps.
At the introduction part: the section comes to fragmentized, the ones that you numbered as subsections could be considered as broad paragraphs in order to see a connection between the pieces. I suggest reformulating the Introduction part taking into account the most important thing here: which is the lacune, the gap in the revised literature that you aim to meet with your work. It should be explicitly stated, underlining the novelty behind it (if there is any!). I suggest considering merging the Introduction part with the Literature review part.
R: the Introduction has been substantially adjusted to reflect these gaps, including the novelty of this article (last two sentences of the before last paragraph of the Introduction). We have removed the subchapters, but we think it would be better to maintain the literature review separated from the introduction as in many articles.
The methodology section should have a subsection about the data. I missed the type of data and the variables used explained clearly.
R: We have introduced a completely new subchapter on the data gathering (4.1) and introduced a new paragraph in chapter 4, before subchapter 4.1.
The section on Results should be formulated clearly about your findings. Things such as Table 8 or the note on page 18 are put there for what reason? DW test table values are known from the researchers, so it seems to me unnecessary to be included. It presents the work in a scholastic language style, not too refined in my opinion.
R: We agreed with removing table 8 (DW test table value), as well as the note on page 18.
Statement such as “…Introduce ADF test here…." on page 9, gives me the idea of an unpolished first version of the manuscript.
R: Removed as it was a mistake.
The results should be aligned also with the line of the journal. Where are your findings located in terms of sustainability issues?
R: we have introduced this alignment in the last 3 sentences of the abstract, first paragraph of the methodology, in the results chapter, above the table 6 and new paragraphs 7 and 8 in the conclusions.
The methodological part: I should express that reading the revisited literature and what you stated in the abstract part: “…doubts have been expressed about the accuracy of econometric models that measure the real impact that banking financial institutions have on boosting the agricultural sector…” I was expecting a very consolidated methodology (in statistical and econometrical terms). Your choice about the econometric model ADL (p, k) (page 9 it is written ADL(p, p), also at the equation 1) is not justified. Why not an autoregressive model AR(p), or a distributed lag model (DL(k)) (selection criteria: serial correlation of residuals, information criteria. significance)?
R: the Abstract has been substantially adjusted and we analyzed the ADL/ARDL model in the literature review chapter. We have added in chapter 3 more detail on the literature used, especially paragraph 6 (Moura), and added paragraphs 9 and 10 with two more literature references (Pham and Nguyen (2020) and Kaleemuddin and Masih (2017)). We have also completely reviewed the methodological chapter to include the rationale of using the ARDL model. The ARDL model has become an important tool for detecting cointegration relationship based on the work of Pesaran and Shin (1999). The authors demonstrate that with an ARDL representation, it is possible to identify cointegration relationships in a system formed by variables that are all I(1), all I(0), or a mixture of stationary variables and variables I(1). This constitutes a great advantage compared to the Johansen cointegration method and even the FMOLS estimator, as both assume tha all variables in the system are I(1). Finally we have correct ADL (p, p) to ADL (p, k).
Then you pass (on page 9 again) from the stationarity attribute of the series to the Granger causality approach and then (page 13) at the ECM approach doubting for cointegration between the two series. Surprisingly I found a simple linear OLS approach on page 14 as the first empirical result. My observation: there is not a well-designed research analysis. My suggestion: There is a clear empirical path to follow: series stationarity -> cointegration -> (if positive) VECM -> FEVD, IRFs and Granger causality; ->(if negative) VAR -> FEVD, IRFs and Granger causality.
R: We have reviewed the path and follow the suggestion in the results chapter.
Moreover: the data is annual, or monthly? I missed it. However, the period of study is a wide period (2003-2022) covering several structural breaks and regime shifts that are not considered. The justification behind this fact? This is not a realistic assumption. All the used techniques (without considering the presence of several structural breaks) provide nonreliable findings and results.
R: data are annual and this was reinforced in the first paragraph of subchapter 4.1. In this paragraph we have also explained why we do not consider any structural breaks. There is political stability in Angola, as a result of the peace achieved after 27 years of post-independence civil war (1975 to 2002). The political regime and party in power from 2003 to 2022 is the same.
Graphical series representations are a bit confusing, and figures and tables lack clearly formulated captions.
R: We have reviewed the graphical series and corrected the paragraph after figure 2.
The work aimed to examine the degree of elasticity of agricultural credit and agricultural growth, in Angola. Nothing about this expression was mentioned or calculated in the Result part. Moreover, are you talking about short or long-run elasticity?
R: We have adjusted paragraph 1 of the results chapter, as well as the paragraph above table 6 (This strong correlation allows for solid sustainable policies in the sense that every monetary unit of kwanza would result, ceteris paribus (keeping everything else constant), in 14.41205 units increase in agricultural GDP.) Furthermore we have this already in the introduction and conclusion chapters.
In general, the topic is interesting but limited in terms of used techniques. I firmly the methodological part of your work is the one that needs to be invested more.
R: the methodological part was substantially reviewed as mentioned above.
Best regards,
Mario and Abreu.

Round 2
Reviewer 3 Report
Dear authors,
I acknowledge every intervention you decided to reflect on your manuscript. I see a positive improvement in terms of content and form.
I remain skeptical about your justification for not considering any regime shift or a structural break in your time series (a structural break does not consist only of the political situation). I repeat what I wrote to you previously: every provided result in the scenario of not considering any structural break is unreliable and statistically irrelevant.
This is a big limitation that: (1) you can try to consider it (instead of all the tests that you used, you should use the respective ones for the case of the presence of structural breaks (i.e. CMR test for time series stationary, TY time domain causality test with dummy variables for structural breaks etc.), (2) you can state it as the major limitation of your work.
Best regards.
Some minor interventions in terms of the language style are needed.
Author Response
Dear Reviewer,
We agree with you on the structural rupture. We have adjusted chapter 7 on Limitations to mention this issue, as it can be seen the text bellow:
"One of the main limitations of this work was that, as data for a longer series were not available, as well as the recognition of structural breaks, there may have been unexplained variability in the data sample; that is, statistical noise may not have been minimized as much as it could have been."
We have also done major language corrections.
We hope that this could address your concern.
Best,
Mario and Abreu.
